# Identifying Those at Risk of Glaucoma: A Deep Learning Approach for Optic Disc and Cup Segmentation and Their Boundary Analysis

**DOI:** 10.3390/diagnostics12051063

**Published:** 2022-04-24

**Authors:** Jongwoo Kim, Loc Tran, Tunde Peto, Emily Y. Chew

**Affiliations:** 1Lister Hill National Center for Biomedical Communications, National Library of Medicine, National Institutes of Health, Bethesda, MD 20894, USA; lotran@mail.nih.gov; 2Centre for Public Health, School of Medicine, Dentistry and Biomedical Sciences, Queen’s University Belfast, Belfast BT7 1NN, UK; t.peto@qub.ac.uk; 3Clinical Trials Branch, Division of Epidemiology & Clinical Applications, National Eye Institute, National Institutes of Health, Bethesda, MD 20892, USA; echew@nei.nih.gov

**Keywords:** glaucoma, fundus image, optic disc, cup, neuroretinal rim, segmentation, fully convolutional network (FCN), mask R-CNN, MAPNet

## Abstract

Glaucoma is a leading cause of irreversible vision loss that gradually damages the optic nerve. In ophthalmic fundus images, measurements of the cup to optic disc (CD) ratio, CD area ratio, neuroretinal rim to optic disc (RD) area ratio, and rim thickness are key measures to screen for potential glaucomatous damage. We propose an automatic method using deep learning algorithms to segment the optic disc and cup and to estimate the key measures. The proposed method comprises three steps: The Region of Interest (ROI) (location of the optic disc) detection from a fundus image using Mask R-CNN, the optic disc and cup segmentation from the ROI using the proposed Multiscale Average Pooling Net (MAPNet), and the estimation of the key measures. Our segmentation results using 1099 fundus images show 0.9381 Jaccard Index (JI) and 0.9679 Dice Coefficient (DC) for the optic disc and 0.8222 JI and 0.8996 DC for the cup. The average CD, CD area, and RD ratio errors are 0.0451, 0.0376, and 0.0376, respectively. The average disc, cup, and rim radius ratio errors are 0.0500, 0.2257, and 0.2166, respectively. Our method performs well in estimating the key measures and shows potential to work within clinical pathways once fully implemented.

## 1. Introduction

Glaucoma is a common visual disorder that can damage the optic nerve and result in vision loss and blindness [1,2,3,4]. Glaucoma is the second leading cause of blindness in the world; the World Health Organization estimated that about 80 million people suffer from this disease [5]. In the USA, about three million people have glaucoma, and it is expected that the number of patients will increase to 4.2 million by 2030 [4]. In general, there is a higher risk of glaucoma for people over the age of 60 years, while African Americans face the risk earlier in life at over 40 years of age [1,4]. Early detection is critical for timely treatment to delay disease progression. Open-angle glaucoma is asymptomatic, and therefore regular eye examinations, including appropriate imaging of the optic disc, are vital for reducing the risk of reduced vision or blindness [6].

Clinical evaluation of structural damage to the optic nerve head is a key to the diagnosis and management of glaucoma [7]. Figure 1 shows an example of a fundus image, where the optic disc (or optic nerve head) is the area where blood vessels and optic nerve fibers enter and exit the eye. The tissue between the cup and optic disc margin is called the neuroretinal rim. Most glaucoma patients with visible signs of the disease have a larger cup compared with the optic disc area, and a non-uniform rim width due to optic nerve damage. Ophthalmologists use the vertical diameter of cup to optic disc (CD) ratio, rim to optic disc (RD) area ratio, Inferior Superior Nasal Temporal (ISNT) area and its thickness, and the ISNT rule as indicators for discriminating between those with and without potential damage [8,9,10]. A CD ratio greater than 0.6 is cause for suspicion of glaucoma [1] and these patients will undergo further testing to establish a diagnosis.

High variability has been noted among ophthalmologists’ decision making based on optic disc images. The average inter-reader agreement for the optic disc is 0.643, and for the cup is even more variable with 0.633 [11]. Correct annotation of the optic disc and cup is time consuming and requires considerable expertise even for trained technicians or clinicians [12].

Several automated computer algorithms have been proposed to segment optic disc and cup areas from fundus images, and to measure the CD ratio automatically. Roychowdhury et al. [13] use morphology to extract bright regions near blood vessels from a fundus image, and then use the Gaussian mixture model to extract the final optic disc from the bright regions. Ellipse fitting is used for post-processing. Morales et al. [14] remove blood vessels from input images using principal component analysis and image enhancement. Then, they segment the optic disc using stochastic watershed transformation. Circular fitting is used in post-processing. Almazroa et al. [15] extract blood vessels using top-hat transformation and Otsu threshold from a fundus image, remove the blood vessels in the image, and segment the cup using fuzzy threshold. Hough transform is applied to estimate the approximate cup boundary in post-processing. Cheng et al. [16] use superpixels, histogram, and statistics surrounding the optic nerve center to segment the optic disc, and superpixels, histogram, statistics, and location information to segment the cup. The Support Vector Machine is used to classify the superpixels, and ellipse fitting is used to finalize the boundaries of the optic disc and cup. The performance of the abovementioned algorithms depends on binarization for estimating the candidate optic discs and cups, and good blood vessel detection algorithms. In addition, ellipse fitting is necessary as post-processing to further improve algorithm performance.

Deep learning has been increasingly applied in medical image processing. Fully Connected Networks (FCNs) (called U-Nets) [17] and their modified architectures are commonly used in the segmentation of medical images. Lim et al. [12] use convolutional neural networks (CNN) to classify every pixel in a region of interest (ROI) in fundus images to classify the optic disc, cup, and other retinal areas. Agrawal et al. [18] use U-shape convolutional networks that have 57 layers and train the networks twice using two different inputs: one uses original images and spatial coordinates, and the other uses two images generated by the Contrast Limited Adaptive Histogram Equalization (CLAHE) operator [19] with different parameters. Ensemble learning (EL) is also used to combine the optic disc and cup segmentation from the two inputs [18]. Sevastopolsky et al. [20] use CLAHE as a preprocessor to improve the input image contrast and a modified U-Net for the segmentation of the optic disc and cup. Fu et al. [21] localize the ROI area and use modified M-shape convolutional networks (called M-Net) and polar coordinates to segment the optic disc and cup. M-Net uses four different sizes of ROIs as inputs to generate four outputs. The final segmentation result is estimated by combining the four outputs. Al-Bander et al. [22] adapts a fully convolutional DenseNet for the segmentation of the optic disc and cup from ROI in a fundus image. They use a post-processing method that selects the largest blob as the optic disc and cup area in the ROI to compensate for segmentation errors in their deep learning model. Yu et al. [23] use modified U-Nets based on ResNet34. Two U-Nets are used: one for ROI detection from a fundus image and the other for optic disc and cup segmentation from the ROI. They use post-morphological processing to select the largest blobs for the optic disc and cup to reduce segmentation errors from the U-Nets. Orlando et al. [24] evaluate twelve different deep learning algorithms for the segmentation of the optic disc and cup from fundus images obtained through the REFUGE challenge. These include networks such as FCNs based on ResNet [25], M-Net [26], Faster R-CNN [27], Mask RCNN [28], and U-Net [17]. Among them, U-Net and Deep LabV3+ [29] show the best performance in optic disc segmentation, and Mask R-CNN shows the best performance in cup segmentation. Four algorithms in their work use EL to improve overall performance.

Most deep learning algorithms use two steps for segmentation. The first step estimates ROI from a fundus image and the next step estimates the optic disc and cup from the ROI. The diameter of the optic disc estimated in the first step is used to crop ROI from a fundus image for the second step. One to three times the optic disc diameter estimated in the first step is commonly used as the width and height of the ROI [18,21,23]. In addition, post-processing algorithms are used after the second step to select the largest blob areas as the final segmentation of the optic disc and cup to alleviate unexpected segmentation errors from their deep learning algorithms [22,23].

A reliable and accurate ROI, where the optic disc is in the center, generates more accurate segmentation results since the segmentation algorithms are trained under the assumption that the optic disc is in the center. The size of the ROI can affect segmentation [30,31]. The optic disc area (pixels) becomes larger than other retina areas when the ROI size is small, and the optic disc area becomes smaller than other retina areas when the ROI size is large. A large ROI includes more geometric information surrounding the optic disc in the ROI. Additionally, a single segmentation algorithm occasionally generates unexpected segmentation errors such as isolated blobs outside of the optic disc in the ROI.

The CD ratio is a measure commonly used to confirm suspicion of glaucoma and monitor progression. However, it only considers vertical diameters of the cup and optic disc. There are other key measures for a suspicious optic disc such as the CD area ratio, RD area ratio, and rim width ratio [8,9,10]. Therefore, analysis of optic disc and cup boundaries is also necessary, especially for the neuroretinal rim area and its thickness. However, few studies have focused on estimating and evaluating this information.

This may be the first step in developing an algorithm for screening for glaucoma. A large proportion of glaucoma remains undiagnosed. It can be a silent disease and often is not diagnosed until the disease is quite severe despite the availability of therapies that can reduce the disease progression and vision loss once glaucoma has been identified. There is already a shortage of physicians with a projected greater shortage in the future. The diagnosis of glaucoma currently requires individual examination by an experienced physician. There is no screening algorithm with high enough sensitivity to identify patients with varying stages of severity of glaucoma.

In this paper, we propose a method to segment the optic disc and cup from fundus images in order to improve the above issues. We adapt Mask R-CNN to estimate accurate and reliable ROIs from fundus images and propose a new segmentation algorithm called the Multiscale Average Pooling Net (MAPNet). The MAPNet uses outputs from three different sizes of convolutional layers (from three Dense Blocks) in DenseNet121 [22] as features and three different sizes of average pooling layers to collect deeper features from the outputs. The MAPNet generates three segmentation outputs from three different sizes of features and adapts an ensemble learning method to combine the segmentation outputs.

The proposed Mask R-CNN allows us to estimate reliable and accurate ROI providing detailed optic disc information such as candidate optic disc mask, ROI coordinates, and probability of optic disc for each candidate mask (or ROI). The proposed MAPNet provides more reliable and consistent segmentation results because it adapts an ensemble learning architecture by combining three segmentation outputs from the three different sizes of features. Based on the segmentation results, we further estimate and evaluate more key measures such as CD area ratio, RD ratio, neuroretinal rim width (for ISNT area and ISNT rules), radius of optic disc, cup, and neuroretinal rim for all directions to provide more clear evidence to confirm suspicion for glaucoma.

The remainder of this paper is organized as follows. Section 2 describes our methods and the dataset used in this experiment in detail. We present the experimental results and discussion in Section 3 and Section 4 and conclude in Section 5.

## 2. Materials and Methods

### 2.1. Datasets

Two publicly available fundus image datasets (REFUGE and RIGA) are used. Among the images in the datasets, 400 images from the REFUGE challenge dataset [24] and 699 images from the RIGA dataset [11] are used in this experiment.

The REFUGE challenge dataset is originally composed of 1200 fundus images from several hospitals and clinics. It is comprised of three sets: 400 training, 400 validation, and 400 test set images. For our experiments, the training set is used since we only obtain annotations of the 400 images. The training set images are comprised of 40 fundus images with glaucoma and 360 fundus images without glaucoma. The images have two different sizes (2124 × 2056 pixels and 1634 × 1634 pixels). Each image has glaucomatous or non-glaucomatous labels generated based on a comprehensive evaluation of the clinical records of patients diagnosed by ophthalmologists. Optic disc and cup annotations are performed by seven ophthalmologists who have an average of eight years of experience, and a majority voting method is used to generate ground truth (GT) annotation results.

The RIGA dataset is collected to facilitate research on computer-aided diagnosis of glaucoma. It is composed of 750 color fundus images from three different sources: MESSIDOR, Bin Rushed, and Magrabi. Among the images in the dataset, 699 fundus images are used in this experiment: 460 MESSIDOR, 145 Bin Rushed, and 94 Magrabi images. The MESSIDOR [32] dataset, also available separately, is originally composed of 1220 fundus color images. The RIGA dataset contains 460 MESSIDOR images having two different image sizes (2240 × 1488 and 1440 × 960 pixels). The Bin Rushed dataset contains 195 images with sizes of 2376 × 1584 pixels. In the dataset, 50 images do not contain fundus images (images without any optic disc and cup annotations). Therefore, we use 145 images from the Bin Rushed dataset. The Magrabi dataset has 95 images; however, we could only download 94 images. The images have a size of 2743 × 1936 pixels. Images from the three sources are obtained from normal and glaucomatous patients. The RIGA dataset contains six types of annotations such as the optic disc area, optic disc centroid, cup area, cup centroid, vertical CD ratio, and horizontal CD ratio. Each fundus image is annotated by six ophthalmologists and the annotations of each ophthalmologist are evaluated by five other ophthalmologists. Unfortunately, high subjectivity has been noted among all the ophthalmologists in the annotations. In the case of optic disc area (and its centroid) annotation, the inter-reader agreement of these peer-to-peer reviews ranges from 0.5985 to 0.6916, and the average agreement is 0.6433. The agreement becomes worse in the case of cup area (and its centroid) annotation, where the inter-reader agreement ranges from 0.5189 to 0.6745, and the average agreement is 0.6325. Since the optic disc and cup annotations are challenging even for ophthalmologists, it is hard to choose GT annotations for the optic disc and cup from among the results of the six ophthalmologists. For our work, we select the results of an ophthalmologist (ophthalmologist #4) who has the best agreement in the optic disc and the results of another ophthalmologist (ophthalmologist #1) who has the best agreement in the cup to use as the GT data for the RIGA dataset.

Table 1 shows the summary of the two datasets used in our experiments. In total, 1099 images are used for the experiment. Figure 2 shows examples from each dataset. Figure 2a–d are from Bin Rushed, MESSIDOR, Magrabi, and REFUGE datasets, respectively. The images in each dataset have different shapes and color intensities.

### 2.2. Proposed Architecture

The fundus images used in our experiments are larger than the input size used in most deep learning segmentation algorithms, impacting memory allocation and resulting in long training times. Despite the resource demands, we use larger images to improve performance. Our proposed method is composed of three steps to alleviate some of the computational demands. Figure 3 shows the architecture of the proposed method. The first step estimates the approximate optic disc and ROI from a fundus image using Mask R-CNN. The second step estimates the optic disc and cup from the ROI using our proposed segmentation algorithm (MAPNet). The third step estimates the CD ratio, CD area ratio, RD area ratio, and neuroretinal rim thickness. The average disc, cup, and rim radius ratio errors are also estimated. The following sections explain each step in detail.

### 2.3. Mask R-CNN for Region of Interest (ROI) Estimation

Our ROI is the area where the optic disc is located in a fundus image. Conventional CNNs require long computational times to detect these ROI areas [30]. Fast R-CNN [33], Faster R-CNN [27], and Mask R-CNN [28] have recently been proposed for object detection fields to speed up the processing time and accuracy. Among them, Mask R-CNN is adapted since it provides segmentation masks (optic disc masks in our case), bounding box coordinates of the segmentation masks (top left and bottom right coordinates of the optic disc bounding boxes), and class probability for each mask (optic disc probability for each mask) as outputs from an input fundus image.

Step 1 in Figure 3 shows the architecture of the Mask R-CNN. The Mask R-CNN uses the Feature Pyramid Network (FPN) to extract feature maps with different scales from multiple layers. Therefore, it generates strong features at various resolution scales compared with single-scale feature map structures. Next, the Region Proposal Network (RPN) scans feature maps in the FPN and proposes possible object regions based on anchors with multiple scales and aspect ratios. The RoI Align technique is applied to keep the size of the feature maps the same using bilinear interpolation so that the same subnet can be used for the next step. Finally, the network identifies the top left coordinates (xtop left, ytop left) and bottom right coordinates (xbottom right left, ybottom right) of the optic disc bounding box and the coordinates are used to estimate several different sizes of ROIs for the following steps. ResNet101 is used as the CNN (backbone network) for the FPN in the proposed network.

Mask R-CNN uses a multi-task loss on each sampled RoI during training as shown in Equation (1).
*L* = *α* × *L_cls_* + *β* × *L_box_* + *γ* × *L_mask_*,(1)
where *L_cls_* is class loss, *L_box_* is bounding box loss (ROI loss), and *L_mask_* is segmentation mask loss. We use equal weights for every loss (*α* = *β* = *γ* = 1.0) in this experiment since class classification, bounding box estimation, and mask segmentation are all equally important.

Figure 4 shows examples of the Mask R-CNN outputs. Figure 4a shows an input image, Figure 4b shows the estimated segmentation mask of the optic disc, and Figure 4c shows the estimated ROI drawn using the estimated coordinates. The coordinates of the estimated bounding box are [(198, 41) and (334, 571)] with the optic disc probability = 0.9999 in this case. Figure 4d shows the estimated optic disc boundary. In Figure 4c,d, green represents the GT annotations and blue represents the estimation results of the Mask R-CNN.

The proposed Mask R-CNN provides three pieces of important information as outputs for the optic disc. Consequently, the network has a large architecture that needs a large amount of computer memory and long training time. To process large images (2124 × 2056 as shown in Table 1) in our datasets, we resize the images to fit our hardware and to reduce computation time. The optic disc area is also small when compared to the whole retinal area, which can affect the accuracy of the optic disc segmentation masks. Therefore, we use the ROI and optic disc segmentation mask of the proposed Mask R-CNN as preprocessing outputs for the next step to have more accurate optic disc and cup segmentation results.

Step 2 in Figure 3 shows the architecture of the proposed segmentation method. It is comprised of two MAPNets, one for optic disc segmentation and the other for cup segmentation. Our proposed method is as follows. We choose a bounding box coordinate (ROI coordinates) having the highest optic disc probability from the proposed Mask R-CNN output for an input fundus image. Using the longest side of either width or height of the bounding box (ROI), we crop an ROI that has 2.0 times of the longest side (in this experiment) from the input fundus image.

Two MAPNet models are trained to segment the optic disc and cup independently. Each MAPNet generates a segmentation output from the ROI, and the output is mapped to the original location in a black background image, which has the same size as the input fundus image.

### 2.4. Multiscale Average Pooling Network (MAPNet) for Optic Disc and Cup Segmentation from ROIs

There are several FCN algorithms using existing convolutional networks as the feature extractor (encoder) to segment images such as DenseNet [22], ResNet [23,25], and PSPNet [34]. The first two models [22,23] use existing deep learning models as the feature extractor (encoder) and adapt decoders that are similar to the architecture of the U-Nets decoder. In the case of the PSPNet, it uses two different types of features: one involves features extracted from existing ResNet models (ResNet50, ResNet101, ResNet152, and ResNet269) and the other involves four global features collected by (global) average pooling since the pooling provides good features for segmentation [34,35]. In the encoding process, deeper layers will generate more sophisticated features. However, it may lose position (geological) information of each feature since the width and height of the feature maps become much smaller compared to the original input image size when encoded layers go deeper. In the decoding process, most segmentation algorithms use several middle layers (similar to the U-Net decoder) to gradually increase the layers’ size to the original input image size. However, the PSPNet does not use the middle layers. Therefore, the upsampling ratio of the PSPNet is higher than other algorithms.

A single FCN usually has good performance; however, it can occasionally generate over/under labeling issues near the optic disc or part of the optic disc [31]. In addition, even though we use the same dataset for training FCNs, all trained FCNs have a different performance since random numbers are used to initialize weights for training. This is a common consistency issue in all machine learning algorithms. Therefore, some algorithms use additional post-processing to choose the largest segmentation blob as the optic disc to resolve these issues [22,23]. The EL method is commonly adapted to segment the optic disc and cup from ROI [18,36]. This method combines the results of multiple FCNs to have more accurate, consistent, and reliable performance than a single machine learning algorithm.

Therefore, we propose a new segmentation algorithm called MAPNet to improve the above issues by adapting the DenseNet121 feature extractor, average pooling-based feature extractor, and an ensemble learning architecture as shown in Figure 5. MAPNet uses three different sizes of features containing two different kinds of information (geological and average feature information). It extracts three different depths and sizes of features from input images to collect geological information using the first, second, and third Dense Blocks in the DenseNet121 feature extractor. It also estimates the average feature information from each piece of geological feature information using three different sizes of average pooling layers. MAPNet uses a different decoding method to generate three independent segmentation outputs directly from the three different sizes of features without using the middle layers. Unlike other DL algorithms that use an ensemble learning as a postprocessor, MAPNet uses an embedded ensemble learning architecture to combine the three segmentation outputs for more robust results.

The proposed model segments the optic disc and cup from ROI images. Input of the proposed model is a 512 × 512 × 3 color image and output is a 512 × 512 × 2 grey level image. The model adapts two feature extractors. The first one modifies the architecture of DenseNet121. The original DenseNet121 is composed of four dense blocks. For the input size 512 × 512 × 3, output size of the first Dense Block is 128 × 128, the second Dense Block has 64 × 64, the third Dense Block has 32 × 32, and the fourth (or last) Dense Block has 16 × 16. We use three Dense Blocks from first to third as shown in the DenseNet-based Feature Extractor (DNFE) in the figure to use three different features having different pieces of geometric and depth information. From the outputs of each of the Dense Blocks, we estimate an additional 384 features using the second feature extractor called the Average Pooling based Feature Extractor (APFE). The APFE is composed of three different sizes of convolution layers that extract three different sizes of average features.

In the decoding process, we use three outputs from the DNFE and another three outputs from the APFE. For example, the output (128 × 128 × 128) of the first Dense Block and output (128 × 128 × 384) of the APFE for the second Dense Block are used to generate a grey level output (512 × 512 × 2) using a convolutional layer (Conv2DTranspose layer). Similar rules are applied for the outputs of the second and third Dense Blocks. Therefore, the MAPNet generates three segmentation outputs from the three different sizes of features (128 × 128 × 512, 64 × 64 × 640, and 32 × 32 × 1408). At the final stage, all three outputs are combined to generate the final output. The final convolutional layer works as the role of ensemble learning to alleviate the over/under segmentation issue and generate a more stable and robust output.

### 2.5. Critical Measures for Screening Potential Glaucomatous Damage

We estimate several measures from the segmentation results such as the CD ratio, CD area ratio, neuroretinal rim to disc (RD) area ratio, and the rim thickness as shown in the Step 3 in Figure 3. The CD ratio is a common measure to classify glaucomatous fundus images as shown in Equation (2). However, the area ratio of the optic disc and cup is also important [8,9,10]. Therefore, we also estimate the CD area ratio and RD area ratio to consider the overall optic disc, cup, and neuroretinal rim areas as shown in Equations (3) and (4). Equations (5)–(7) show formulas to estimate errors between GT and estimated CD, CD area, and RD area ratios, respectively.
(2)Cup to Disc Ratio (k)=CDR(k)=Vertical Diameter of Cup(k)/Vertical Diameter of Optic Disc(k)
(3)Cup to Disc Area Ratio (k)=CDAR(k)=Total Number of Pixels in Cup(k)/Total Number of Pixels in Optic Disc(k)
(4)Rim to Disc Area Ratio (k)=RDAR(k)=Total Number of Pixels in Rim(k)/Total Number of Pixels in Optic Disc(k)
(5)Cup to Disc Ratio Error(k)=CDRE(k)=|GT CD Ratio(k)−Estimated CD Ratio(k)|  
(6)Cup to Disc Area Ratio Error(k)=CDARE(k)=|GT CD Area Ratio(k)−Estimated CD Area Ratio(k)|  
(7)Rim to Disc Area Ratio Error(k)=RDARE(k)=|GT RD Area Ratio(k)−Estimated RD Area Ratio(k)|  

Since the rim area and thickness are critical measures to find notches from fundus images for screening glaucomatous fundus damage, the accuracy of the optic disc and cup boundaries is important for the estimation. However, there have been no observed studies that have attempted to estimate and analyze the rim boundaries. Therefore, we measure them in all angles for the first time to our knowledge. These measures are necessary to estimate the rim thickness and rim areas of each direction (ISNT direction) or to apply the ISNT rule (Inferior > Superior > Nasal > Temporal) to diagnose retinal conditions later. Figure 6 shows a detailed explanation for measuring the accuracy of the optic disc and cup boundaries, and the rim thickness. In Figure 6a, (*x_c_*, *y_c_*) is the centroid of the optic disc, *m* is angle between the horizontal line (white line) and a line passing from the centroid to a point on the optic disc boundary (*x_m_*, *y_m_*). *Rim radius* (*m*) means *Optic Disc Radius* (*m*) minus *Cup Radius* (*m*) at angle *m*. We measure the optic disc, cup, and neuroretinal rim radiuses of all 360° for all test images, increasing *m* by one degree in each step.

Figure 6b,c show examples of optic disc and cup boundaries of GT and our estimated results, respectively. Figure 7 shows the radiuses estimated from Figure 6b,c. Figure 7a shows GT and our estimated optic disc radiuses. In the graph, green represents the GT radius, blue represents our estimated radius, and red represents the absolute difference (error) of the two radiuses. There is a noticeable error at 150° (between Superior and Nasal) with a ten-pixel difference. Figure 7b shows GT and our estimated cup radiuses. There is a noticeable error at 315° (between Inferior and Temporal) with an eleven-pixel difference. Figure 7c shows GT and our estimated rim radiuses. Since there are noticeable errors in the optic disc at 150° and cup at 315°, the rim also shows a twelve-pixel difference error at 150° and nine-pixel difference error at 315°. Figure 7d shows three error graphs of the optic disc, cup, and neuroretinal rim collected from Figure 7a–c. Since images used in the four different datasets have different image sizes, each image has different pixel sizes of the GT optic disc and cup radiuses. Therefore, absolute errors as shown in Figure 7d may not be a good measure to evaluate and compare the boundary errors of all images in the datasets. Therefore, relative errors are also used to evaluate the errors. Equations (8)–(10) are used to estimate the relative error (ratio error) of the optic disc, cup, and rim radiuses at angle m, respectively. In the equations, GT radiuses are used as the denominator to normalize the errors. Figure 7e shows the ratio errors. Since our estimated results show better performance in the optic disc than the cup, and the GT radius of the optic disc is about two times larger than those of the cup and rim, the optic disc ratio error (blue) shows a lower value in most ranges compared to those of the cup (red) and rim (green).

To evaluate the average radius accuracy of the optic disc, cup, and neuroretinal rim, we also estimate their errors in all 360° angles and average them as shown in Equations (11)–(13).
(8)$$Disc\;Radius\;Ratio\;Error\left(m\right)=DRRA\left(m\right)=\frac{\left|GT\;Disc\;Radius\left(m\right)-Estimated\;Disc\;Radius\left(m\right)\right|}{GT\;Disc\;Radius\left(m\right)}$$
(9)$$Cup\;Radius\;Ratio\;Error\left(m\right)=CRRA\left(m\right)=\frac{\left|GT\;Cup\;Radius\left(m\right)-Estimated\;Cup\;Radius\left(m\right)\right|}{GT\;Cup\;Radius\left(m\right)}$$
(10)$$Rim\;Radius\;Ratio\;Error\left(m\right)=RRRE\left(m\right)=\frac{\left|GT\;Rim\;Radius\left(m\right)-Estimated\;Rim\;Radius\left(m\right)\right|}{GT\;Rim\;Radius\left(m\right)},$$
where Rim Radius(m)=|Disc Radius(m)−Cup Radius(m) |
(11)$$Average\;\;Disc\;Radius\;Ratio\;Error\left(k\right)=ADRRE\left(k\right)=\left[\;\overset{360}{\underset{m=0}{\sum }}\frac{\left|GT\;Disc\;Radius\left(m\right)-Estimated\;Disc\;Radius\left(m\right)\right|}{GT\;Disc\;Radius\left(m\right)}\;\right]/\;360$$
(12)$$Average\;Cup\;Radius\;Ratio\;Error\left(k\right)=ACRRE\left(k\right)\;=\left[\;\overset{360}{\underset{m=0}{\sum }}\frac{\left|GT\;Cup\;Radius\left(m\right)-Estimated\;Cup\;Radius\left(m\right)\right|}{GT\;Cup\;Radius\left(m\right)}\;\right]/\;360$$
(13)$$Average\;Rim\;Radius\;Ratio\;Error\left(k\right)=ARRRE\left(k\right)\;=\left[\;\overset{n=360}{\underset{m=0}{\sum }}\frac{\left|GT\;Rim\;Radius\left(m\right)-Estimated\;Rim\;Radius\left(m\right)\right|}{GT\;Rim\;Radius\left(m\right)}\;\right]/\;360$$

## 3. Results

We use two datasets, RIGA and REFUGE, which contain 1099 images and use ten-fold cross-validation to generate ten test sets. The first set contains 987 images for training and 112 images for testing. The images in each dataset are also assigned to each fold equally after sorting the images using random numbers. Since most deep learning algorithms use a square shape as the input, we crop the original fundus images to make a square shape as shown in Figure 2.

Several metrics are used to evaluate our optic disc and cup segmentation performance such as Jaccard Index (JI), Dice Coefficient (DC), Sensitivity, Specificity, and Accuracy as shown in the equations from Equation (14) to Equation (18). TP, FN, FP, and TN mean true-positive, false-negative, false-positive, and true-negative, respectively.
(14)$$Jaccard\;Index\;\left(JI\right)=\frac{\mathrm{TP}}{\mathrm{TP}+\mathrm{FP}+\mathrm{FN}}$$
(15)$$Dice\;Coefficient\;\left(DC\right)=\frac{2\times \mathrm{TP}}{2\times \mathrm{TP}+\mathrm{FP}+\mathrm{FN}}$$
(16)$$Sensitivity\;\left(SEN\right)=\frac{\mathrm{TP}}{\mathrm{TP}+\mathrm{FN}}$$
(17)$$Specificity\;\left(SPE\right)=\frac{\mathrm{TN}}{\mathrm{TN}+\mathrm{FP}}$$
(18)$$Accuracy\;\left(ACC\right)=\frac{\mathrm{TP}+\mathrm{TN}}{\mathrm{TP}+\mathrm{TN}+\mathrm{FP}+\mathrm{FN}}$$

Python and TensorFlow’s Keras APIs [37,38] are used to implement the proposed model. The hardware configuration used for this experiment is 2 × Intel Xeon Gold 5218 processors 2.3 GHz, 64 hyper-thread processors, 8 × RTX 2080 Ti, and Red Hat Enterprise Linux 7.

### 3.1. ROI Detection Using Mask R-CNN

Mask R-CNN is used to segment the optic disc from the fundus images to estimate the ROI. To train the model, we augment each training image into ten using a vertical flip and five rotations (−20°, −10°, 0°, 10°, and 20°) to increase the number of images in the training dataset. Since ten-fold cross-validation is adapted, ten training/test sets are used in this experiment. In each training set, 80% of the images are used for training and 20% are for validation. Input images are reshaped to a square shape (1200 × 1200). We use ResNet101 as a backbone, five backbone strides (4, 8, 16, 32, and 64), epochs = 100, batch size = 1, learning rate = 0.001, learning rate2 = 0.0001, learning momentum = 0.9, and weight decay = 0.001 to train the network. For the Region Proposal Net (RPN), we use three RPN anchor ratios (0.5, 1.0, and 2.0), five RPN anchor scales (128, 192, 256, 184, and 512), and the RPN non-max suppression (NMS) threshold = 0.7. Equal weights are used for the Mask R-CNN class, Mask R-CNN bounding box, and Mask R-CNN mask losses for the optimization of the model. Figure 8 shows the results using images from four different datasets. Green is GT data and blue is the proposed Mask R-CNN output. The top row shows outputs of the optic disc boundary, and the bottom row shows ROI outputs. Although there is a small discrepancy between the GT data and the Mask R-CNN outputs, the outputs show relatively accurate results. Overall, the proposed Mask R-CNN shows 0.9037 JI, 0.9489 DC, 0.9231 Sensitivity, 0.9996 Specificity, and 0.9983 Accuracy.

The Mask R-CNN has complex architecture and is computationally expensive in training compared with other deep learning segmentation algorithms. However, this step is necessary to estimate reliable and consistent ROIs from all input images for accurate optic disc and cup segmentation. Wrong ROI results in this step cannot provide any chance in the next segmentation step. Since the proposed Mask R-CNN provides the probability of each class for each candidate ROI (mask) as one of their outputs, the real optic disc ROI (mask) can be selected easily even in the situation of over labeling. From this point of view, the Mask R-CNN is a more reliable method in our experiments. Therefore, the Mask R-CNN is adapted for the ROI detection.

### 3.2. Training the Proposed MAPNet

Two MAPNets are used to segment the optic disc and cup from an ROI independently. The input size of the MAPNet is 512 × 512 × 3 color images and outputs are 512 × 512 × 2 images as shown in Figure 5. We augment each ROI into ten using a vertical flip and five rotations (−10°, −5°, 0°, 5°, and 10°) to increase the number of images in the training dataset. In each training set, 80% of the images are used for training and 20% for validation. We first use a pretrained DenseNet121 weight from ImageNet [39]. However, the results using transfer learning do not show good performance. Therefore, we train all layers in the proposed model (encoder and decoder) to improve the performance. We use a maximum of 100 epochs, batch size = 20, Adam optimization algorithm (learning rate = 10^−4^, first beta = 0.9, second beta = 0.999, epsilon = 10^−7^, and decay = 0.0) for training, and save the best training results as an output during the training time.

Since the images used in our experiments are from four different datasets, the size of images and distribution of color intensities are different as shown in Figure 2. To normalize the variation in color intensity, the mean and standard deviation of each image are used before training as shown in Equation (19).
(19)$$I_{N}\left(i,j,k\right)=\frac{I\left(i,j,k\right)-m_{k}}{\sigma_{k}},$$
where *I*(*i*, *j*, *k*) is an input image, *I_N_*(*i*, *j*, *k*) is a normalized input image of *I*(*i*, *j*, *k*), *i* and *j* are coordinates of the images *I* and *I_N_*, *k* = Red, Green, or Blue channel, $m_{k}$ is the mean of pixel values of channel *k*, and *σ_k_* is the standard deviation of pixel values of channel *k*.

Figure 9 shows six ROI images and their corresponding ground truth labels of the optic disc and cup used in this experiment. The first row shows ROI images (ROI size = 2.00 times of the longest side of optic disc diameter). The second and third rows are for ground truth labels of the optic disc and cup, respectively. All images are resized to 512 × 512 for the MAPNet. The ROI images are cropped from the Mask R-CNN outputs of fundus images.

Several different sizes of ROIs are tried to find the best sizes for the segmentation. Table 2 shows the optic disc segmentation results of MAPNet for four different ROI sizes (1.25, 1.50, 1.75, and 2.00 times of the longest side of optic disc diameter) for a dataset. A small ROI has more information (pixels) for optic disc and cup, but less information (pixels) for other retina areas. A large ROI has less information for the optic disc and cup, but more information for other retina areas. The MAPNet using ROI size = 2.00 shows the best performance in JI (0.9355) and DC (0.9665). Since JI and DC are the most important metrics in image segmentation, we choose ROI size = 2.00 as the best size for the proposed MAPNet.

### 3.3. Segmentation of the Proposed MAPNet

Figure 10 shows the segmentation results of the proposed method. In output images, green represents the GT annotation and blue represents the results of the proposed method. The first column (a) shows input fundus images. The second and third columns (b and c) show outputs of the proposed method for optic disc and cup segmentation, respectively. The magnified images of the optic disc output (b) are shown in column (d) and the magnified images of the cup segmentation output (c) are shown in column (e). The boundaries of the GT annotations and the results of the proposed method are very close to each other.

Figure 11 shows more optic disc and cup segmentation outputs of the proposed method. Two images are collected from each of the four datasets. All images in the figure are obtained after magnifying the ROI areas of the proposed model outputs. In the images, green and blue have the same meaning as in Figure 10. The first rows are the optic disc segmentation results, and the second rows are the cup segmentation results. In the first and second rows, the two images in the same column are the results from the same fundus image. The optic disc segmentation result of (a) (first row) has 0.9373 JI and 0.9677 DC, and the cup segmentation result of (a) (second row) has 0.8691 JI and 0.9300 DC. The optic disc segmentation result of (b) has 0.9452 JI and 0.9718 DC and the cup segmentation result of (b) has 0.9115 JI and 0.9537 DC. Among all the results, the optic disc segmentation result of (e) shows the best accuracy with 0.9807 JI and 0.9902 DC, and the cup segmentation result of (g) shows the best accuracy with 0.9333 JI and 0.9655 DC.

#### 3.3.1. Optic Disc Segmentation

Table 3 shows a comparison of the proposed optic disc segmentation results with other methods. The second column shows methods used by other developers. Among the methods, two algorithms use n-fold cross validation to verify their performances, but five do not. “[4-fold CV]” in the first row means four-fold cross-validation is used for evaluation and “[1-fold]” in the second row means no cross-validation is used. The third column shows datasets used for training and/or testing. The fourth column is the number of images (Train Images) used to train, the fifth column (Test Images) shows the number of images used to test, and the metrics estimated from the test results are shown in columns six and seven. In the case of methods using n-fold cross-validation, numbers in the columns of Train Images and Test Images are the same since all images are used for training and testing at least once. Several methods estimate their performances after independent training and testing for each database. These performances have different metric values for each dataset. Therefore, we estimate the average (weighted average) of all datasets using the numbers in Test Images as weights as shown in Equation (20).
(20)$$Average=\sum_{i}metric\left(i\right)\times n\left(i\right)/\sum_{i}n\left(i\right),$$
where *metric* (*i*) is the metric estimated from the dataset *i*, and *n*(*i*) is the number of Test Images in the dataset *i*.

Lim et al. [12] have 0.8780 JI for 1200 MESSIDOR images and 0.9157 JI for 235 SEED DB images. Therefore, the average JI becomes 0.8842 ((0.8780 × 1200 + 0.9157 × 235)/1435 = 0.8842) as shown in the table. Agrawal et al. [18] train EL models and have 0.8800 DC from 40 REFUGE images. Yu et al. [23] train modified U-Nets using two datasets (MESSIDOR and Bin Rushed). Then, they fine tune the U-Nets in each dataset using 50% of DRISHTI images and 80% of RIM ONE images. Their average results show 0.9410 JI and 0.9694 DC. Orlando et al. [24] show the results of the top three teams from the REFUGE challenge: 400 images are used for training, another 400 images for validation, and the other 400 images for testing. The three teams which have the best performance are shown in the table, and the best DC is 0.9602.

We test deep learning algorithms based on Modified U-Nets and/or Mask R-CNN to compare them with the proposed MAPNet. We use the REFUGE and RIGA (MESSIDOR, Bin Rushed, and Magrabi) datasets and ten-fold cross-validation (ten training/testing sets) to evaluate our performance as shown in rows eight to ten. Unlike other methods, we make one overall dataset by combining the four datasets and use it to train our models to make more general and reliable methods for fundus images in any datasets. Since ten-fold cross-validation is adapted in our experiment, our result is estimated by averaging the results of ten (training/test) sets.

In the three results of Kim et al. [31], we use modified U-Nets that we previously developed: U-Net2 for two class classification and U-Net3 for three class classification. The first two results use Modified U-Net3 or Mask R-CNN to segment the optic disc from the fundus images. The third result uses Mask R-CNN to crop the ROIs from the fundus images and Modified U-Net2 to segment the optic disc from the ROIs. Among the three results, the result using Mask R-CNN and Modified U-Net2 shows better performance with 0.9234 JI and 0.9597 DC than other algorithms.

The last two rows (Proposed Method) show the performance of our proposed method. We also use the same datasets and ten-fold cross-validation method (used in the results of Kim et al. [31]) to evaluate our performance. The first row among the two is the result estimated by averaging the results from the ten training/testing sets and the second row is the best result among the ten sets. The average result shows 0.9381 JI and 0.9679 DC and the best result shows 0.9432 JI and 0.9707 DC.

The proposed method shows good performance compared with other methods. It is hard to compare the performance of each method since each uses different datasets and different evaluation methods. Our best result shows the best performance and our average result shows the second-best performance. The performance from Yu et al. [23] comes from a small test data without using cross-validation. For example, in the case of RIM-ONE results, 400 MESSIDOR, 195 Bin Rushed, and 97 RIM-One images are used for training, and 32 RIM-ONE images are used for the test. Since the performance of our method is estimated from the largest data (1099 images) from four different datasets, we expect that it may produce more reliable, robust, and consistent results for fundus images from other datasets.

#### 3.3.2. Cup Segmentation

Table 4 shows a comparison of cup segmentation results. The column information in the table is the same as in Table 3. Equation (20) is also used to estimate the average of each metric for the table. In the second row, Agrawal et al. [18] use EL methods and two datasets for training and have 0.6400 DC from 40 REFUGE images. Yu et al. [23] use two datasets (MESSIDOR and Bin Rushed) for training a CNN. Then, they fine tune the CNNs in each dataset using 50% of DRISHTI images and 80% of RIM ONE images. Their average results show 0.7973 JI and 0.8725 DC. Orlando et al. [24] show results of the top three teams from the REFUGE challenge: 400 images are used for training, another 400 images for validation, and the other 400 images for testing. The top three teams which have the best performance are shown in the table, and the best DC is 0.8837.

We test several deep learning algorithms based on Modified U-Nets and/or Mask R-CNN as shown in the three results of Kim et al. [31]. We also use the same datasets and ten-fold cross-validation method used for the optic disc segmentation to evaluate our performance. The results of the Modified U-Net3 and Mask R-CNN are estimated from the fundus images. The third result uses Mask R-CNN to crop ROIs from fundus images and Modified U-Net2 to segment the cup from the ROIs. Among the three results, the result using Mask R-CNN and Modified U-Net2 shows the best performance with 0.7833 JI and 0.8742 DC.

The last two rows (Proposed Method) show the performance of our proposed method. The first row among the two is the result estimated by averaging the results from the ten training/testing sets and the second row is the best result among the ten sets. The average result shows 0.8222 JI and 0.8996 DC, and the best result shows 0.8355 JI and 0.9082 DC. The proposed method shows the best performance. Unlike other algorithms [22,23] that use post-processing to improve the output of their deep learning algorithms, our proposed method produces robust results without using post-processing for the results since it produces results from three outputs from three different sizes of features.

### 3.4. Key Measures Estimated from the Proposed Ensemble Learning Results

We estimate several important measures from the proposed segmentation results to confirm suspicion of glaucoma: cup to disc ratio, cup to disc area ratio, neuroretinal rim to disc ratio, disc radius, cup radius, and neuroretinal rim radius (thickness). Then, we compare them to the measures estimated from the GT data. Equations (2)–(13) (in Section 2.5) show formulas to estimate the difference (error) of the two measures. Among the measures, the neuroretinal rim radius (Figure 7c) is one of the key measures since we can estimate the neuroretinal rim area, and the ISNT rules can be applied from the information in the graph.

Table 5 shows the mean error of the measures estimated from the proposed model outputs. We estimate each measure for each dataset and the whole dataset. We could not compare the measures with other algorithms since few studies use similar measures for the evaluation, and the formulas for the neuroretinal rim radius (thickness) and radiuses of the optic disc and cup are proposed for the first time to our knowledge. The cup to disc ratio error (CDRE) in the first row shows 0.0451 in total. This means that the proposed algorithms show over 0.9549 accuracy. Sun et al. [40] also measured the CDRE for Drishti-GS and RIM-One datasets, which showed 0.0499 and 0.0630. All CDRE errors of the four datasets from the proposed model show less error values than the two errors.

The cup to disc area ratio error (CDARE) in the second row shows 0.0376; this ratio is better than CDRE as the area ratio considers all areas of optic and disc. The rim to disc area ratio error (RDARE) in the third row has the same performance as CDARE. The fourth to sixth rows show errors related to the optic disc, cup, and rim radiuses. The average disc radius ratio error (ADRRE) in the fourth row shows 0.0500 in total, which means the proposed algorithms show 0.9500 accuracy in the optic disc boundary estimation. The average cup radius ratio error (ACRRE) in the fifth row shows a relatively high error rate of 0.2257 when compared to the error of the optic disc (ADRRE). This demonstrates that cup boundary estimation is more challenging work compared to optic disc boundary estimation. The average rim radius ratio error (ARRRE) in the sixth row shows 0.2166 in total. The error ratio is higher than the error of the optic disc (ADRRE) but less than the error of the cup (ACRRE). Overall, the proposed algorithms show robustness in CDRE, CDARE, RDARE, and ADRRE but less accuracy in ACRRE and ARRRE. Therefore, further studies are needed to improve segmentation accuracy, especially in the cup, to have more accurate measures and to screen suspicious glaucoma based on the measures.

We analyze our results further using histograms as shown in Figure 12. Figure 12a shows the histogram of the estimated CD ratio error. The horizontal axis shows the interval for the error, and the vertical axis shows the count for each range. As shown in the table, 44.22% of the results have less than 3% error (first three bars in error range [0.00, 0.03]), and 94.27% of the results have less than 10% error (error range [0.00, 0.10]). This shows that the proposed algorithm produces reliable results in CD ratio estimation. Figure 12b shows the histogram of the estimated CD area ratio error. As shown in the table, 53.87% of the results have less than 3% error (first three bars in error range [0.00, 0.03]), and 94.81% of the results have less than 10% error (error range [0.00, 0.10]). The proposed algorithm also shows reliable results in the CD area ratio estimation. When we compare the histograms of CDRE and CDARE, CDARE shows slightly better performance since it considers all data (pixels) instead of considering pixels in the vertical direction.

Figure 13 shows the histograms of the average optic disc, cup, and neuroretinal rim radius ratio errors. The histogram for the optic disc (Figure 13a) shows that 90.54% of the results are within the 5% error range ([0.00, 0.05]). In the case of the cup shown in Figure 13b, 90.01% of the results are within the 20% error range ([0.00, 0.20]). In the case of the neuroretinal rim shown Figure 13c, 84.80% of the results are within the 20% error range and 91.72% of the results are within the 25% error range ([0.00, 0.25]). The histograms show higher accuracy in the estimation of the optic disc boundary than the cup or rim boundaries.

## 4. Discussion

We propose a deep learning method to segment the optic disc, cup, and neuroretinal rim to resolve the inter-reader variability issue of human annotators. Mask R-CNN is adapted to estimate the ROIs from the fundus images and MAPNet is proposed for the segmentation of the optic disc and cup from the ROIs. The proposed MAPNet uses outputs of the first three Dense Blocks of DenseNet121 and three APFEs for feature extractors and combines three segmentation outputs to estimate the final robust results. The proposed method analyzes the accuracy of neuroretinal rim thickness, and optic disc and cup boundaries for the first time to our knowledge.

Several important issues are observed for the segmentation and analysis of the optic disc and cup. Through our empirical experiments, we find that accurate and reliable ROI detection improves the optic disc and cup segmentation accuracy. We compare the proposed Mask R-CNN with an FCN with U-Net architecture (Modified U-Net) in ROI detection (optic disc segmentation). In both deep learning methods, the smallest rectangle covering the estimated optic disc output becomes an ROI output of each method. Therefore, both optic disc outputs are compared with the GT optic disc mask. The FCN shows a slightly better performance (JI = 0.9068 and DC = 0.9499) overall than the proposed Mask R-CNN. However, the FCN has issues when it generates unexpected extra masks (over labeling) as the output of the optic disc. The FCN only provides optic disc masks as outputs. Therefore, extra post-processing is needed to find the real optic disc mask (ROI) and to remove other over labeled masks. The FCN can generate more issues for choosing the real ROI, especially when large drusen or lesions are over labeled as ROIs. However, the proposed Mask R-CNN provides candidate ROIs (optic disc masks) and the probability of the optic disc for each ROI (mask) as outputs. We choose the ROI with the highest probability as the real ROI output. Therefore, the proposed Mask R-CNN provides more reliable ROIs for the next step, which estimates the real optic disc and cup from the ROIs using the proposed methods.

The proposed MAPNet is used to segment the optic disc and cup from the ROIs as the next step. We find that ROI size also affects the segmentation accuracy since different size ROIs include different geometric information of fundus images. Four different sizes of ROIs (1.25, 1.50, 1.75, and 2.00 times of the longest side of optic disc diameter) are evaluated and the ROI size = 2.00 provides the best optic disc segmentation results for the proposed model. Cup segmentation is a more challenging issue than optic disc segmentation since the boundary between the optic disc and cup is more ambiguous than that of the optic disc and other retina areas.

The MAPNet for three classes (optic disc, cup, and other classes) is also implemented to segment the optic disc and cup simultaneously. However, results from the MAPNet for two classes show better performance than the results from MAPNet for three classes. Therefore, MAPNet for two classes is adapted in the proposed method.

Designing the decoding part of the proposed MAPNet is another challenging issue. Since occasional over/under labeling is a common issue when using a single FCN algorithm, ensemble learning is frequently adapted to alleviate the issue. Therefore, the proposed model generates three outputs from three different types/sizes of features and combine the three outputs to estimate the final segmentation results.

In addition to the optic disc to cup ratio, optic disc to cup area ratio, neuroretinal rim to disc area ratio, neuroretinal rim thickness, and optic disc and cup boundaries (radiuses) are critical measures to confirm suspicion of glaucoma. Since JI and DC do not correlate with the accuracy of the optic disc and cup boundaries, we propose formulas to analyze the accuracy of the neuroretinal rim, optic disc, and cup radiuses of all directions for the first time to our knowledge. These will enable us to estimate the lengths and areas of inferior, Superior, Nasal, and Temporal, minimum neuroretinal rim distance, and to apply ISNT rules to find notching in the fundus images. There are limits to this study. The collection of reliable fundus image annotations is challenging due to high inter-reader variability. Therefore, annotations from several experienced experts are necessary to develop and improve the performance of the proposed deep learning models. Additionally, the number of images in the dataset is still small compared to the datasets used in other deep learning applications. Thus, the proposed algorithms may have generalization and robustness issues since the dataset does not cover a diverse range of fundus images and patterns of the optic disc and cup. A large-scale collection of fundus images is necessary to make more robust deep learning models.

Studies on other measures are also necessary for using different modalities such as post-illumination pupil response [41], swinging flashlight test [42], and pupillary signal [43]. Combined analysis of these measures with our key measures will help to improve the diagnosis of glaucomatous features more accurately.

Optic disc and cup annotation requires the judgment of professional graders and is considered time-consuming work. It requires about eight minutes per eye based on the Klein protocol [44]. The proposed method needs about 13.37 s on average per fundus image estimating all the key measures including the optic disc and cup segmentation. Among the time, 10 s is for ROI detection and the remaining 3.37 s is for the segmentation of the optic disc/cup and estimation of the key measures. The proposed method adapts Mask R-CNN for the ROI detection to focus more on accuracy than processing time. The processing time can be further improved by adapting other DL algorithms for the detection. When comparing the processing time of the grader and the proposed method, the proposed method is much faster than professional graders. Therefore, this shows a potential benefit for using the proposed method if it is fully implemented and is used in the real field.

## 5. Conclusions

This paper proposes an automatic method to segment the optic disc and cup from fundus images using deep learning algorithms. We also propose several measures to evaluate segmentation results and to use as key factors for the diagnosis of glaucomatous features. The method comprises three steps. A Mask R-CNN is adapted to estimate the ROI from a fundus image. The proposed MAPNet is used for the segmentation of the optic disc and cup from the ROI. The cup to disc ratio, cup to disc area ratio, neuroretinal rim to disc area ratio, optic disc radius, cup radius, and neuroretinal rim radius are estimated as the last step. The MAPNet uses three DNFEs and three APFEs for feature extraction (encoding process) and uses an ensemble learning architecture in the decoding process to combine three segmentation results.

We evaluate the performance of our proposed method, using four datasets and ten-fold cross-validation. The proposed method shows 0.9381 JI and 0.9679 DC in optic disc segmentation, and 0.8222 JI and 0.8996 DC in cup segmentation. We have a 0.0451 error on average for CD ratio estimation, 0.0376 error on average for CD area ratio estimation, and 0.0376 error for neuroretinal RD area ratio estimation. We also analyze the boundaries of the optic disc, cup, and neuroretinal rim for the first time to our knowledge, and have 0.050 for the optic disc radius ratio error, 0.2257 for the cup radius ratio error, and 0.2165 for the cup radius ratio error.

The proposed method demonstrates the effectiveness in the segmentation of the optic disc and cup when compared with other state-of-the-art methods and provides several key measures to screen potential glaucomatous damage. It also shows the potential that the proposed method can work as a second reader for ophthalmologists. We plan to explore different FCNs and hand-made features to further improve the performance of the proposed method.

## Figures and Tables

**Figure 1 diagnostics-12-01063-f001:**
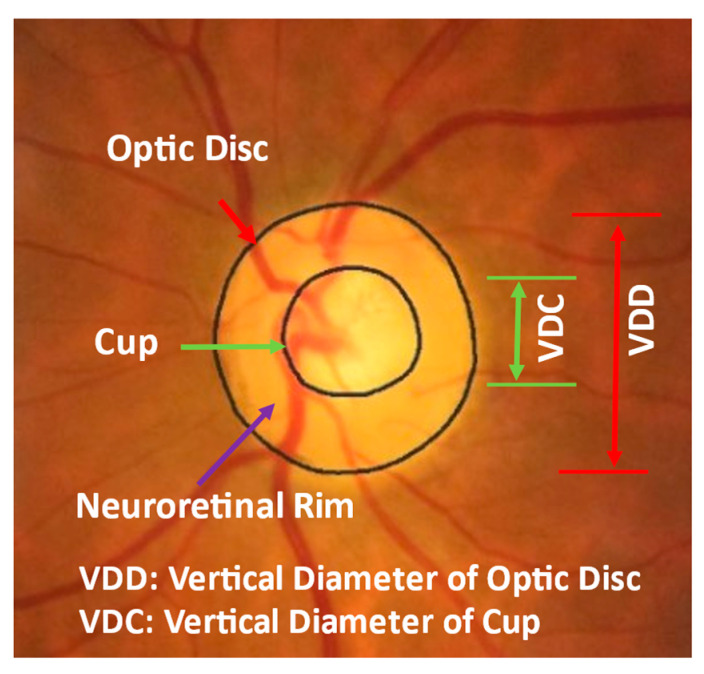
Optic disc, cup, and neuroretinal rim in a fundus image. VDD is vertical diameter of optic disc and VDC is vertical diameter of cup.

**Figure 2 diagnostics-12-01063-f002:**
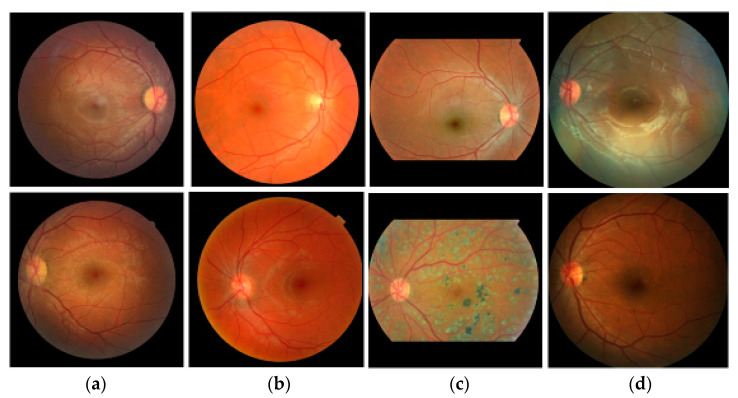
Examples of fundus images. (**a**) Bin-Rushed (RIGA), (**b**) MESSIDOR (RIGA), (**c**) Magrabi (RIGA), (**d**) REFUGE.

**Figure 3 diagnostics-12-01063-f003:**
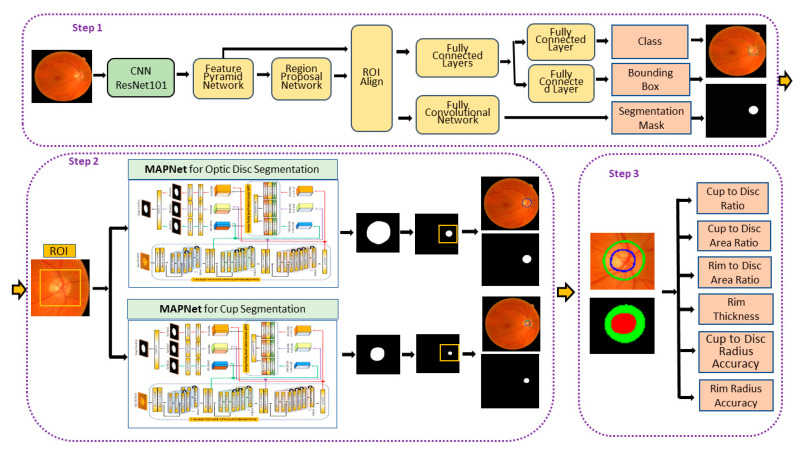
The architecture of the proposed algorithms. Step 1 crops a ROI from an input fundus image using Mask R-CNN. Step 2 segments optic disc and cup areas from the ROI using the proposed segmentation algorithm called MAPNet. Step 3 estimates the key measures from the segmentation results to screen for potential glaucomatous damages in the fundus images. Detail architecture of the MAPNet is shown in the Section 2.4.

**Figure 4 diagnostics-12-01063-f004:**
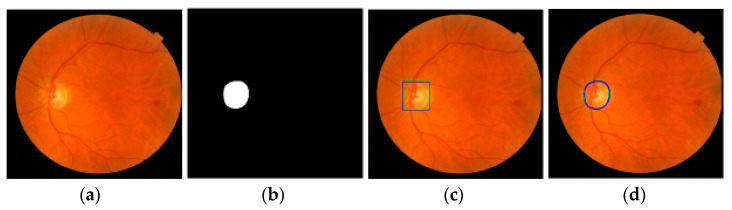
Mask R-CNN results. (**a**) Original input image. (**b**) The results of the Mask R-CNN. (**c**) GT and estimated ROIs. (**d**) GT and estimated optic disc boundaries. Green is ground truth and blue is the estimation results.

**Figure 5 diagnostics-12-01063-f005:**
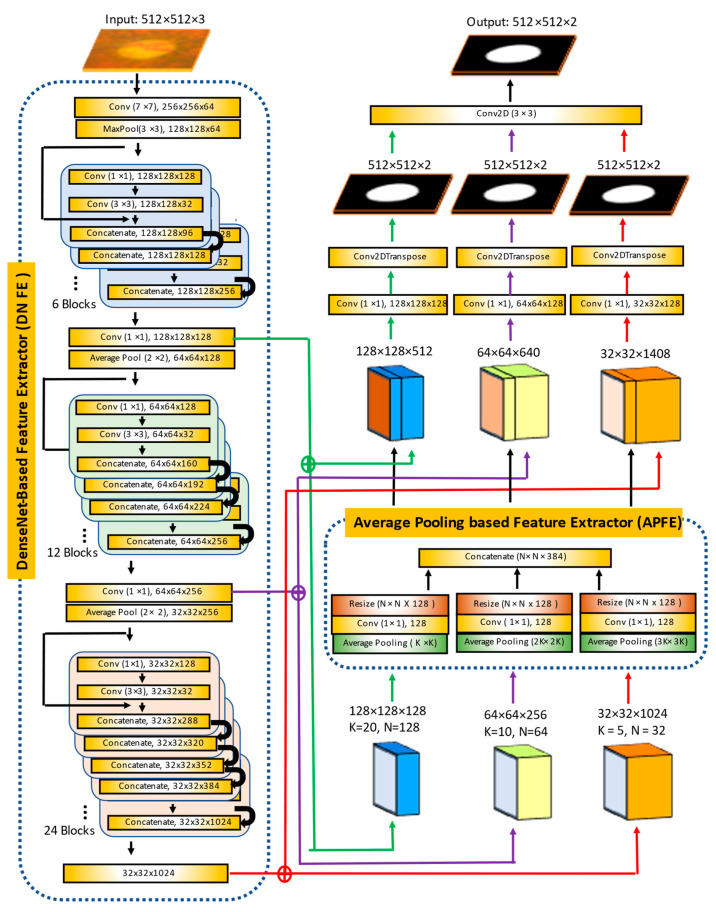
Architecture of the proposed MAPNet used in this experiment.

**Figure 6 diagnostics-12-01063-f006:**
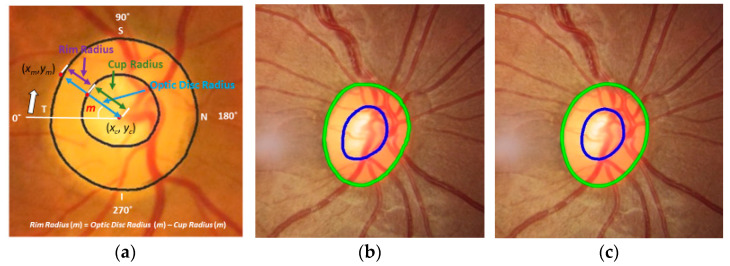
(**a**) Definition of optic disc, cup, and neuroretinal rim radiuses. (**b**) GT optic disc and cup boundaries. (**c**) Estimated optic disc and cup boundaries. Green means optic disc boundary and blue means cup boundary.

**Figure 7 diagnostics-12-01063-f007:**
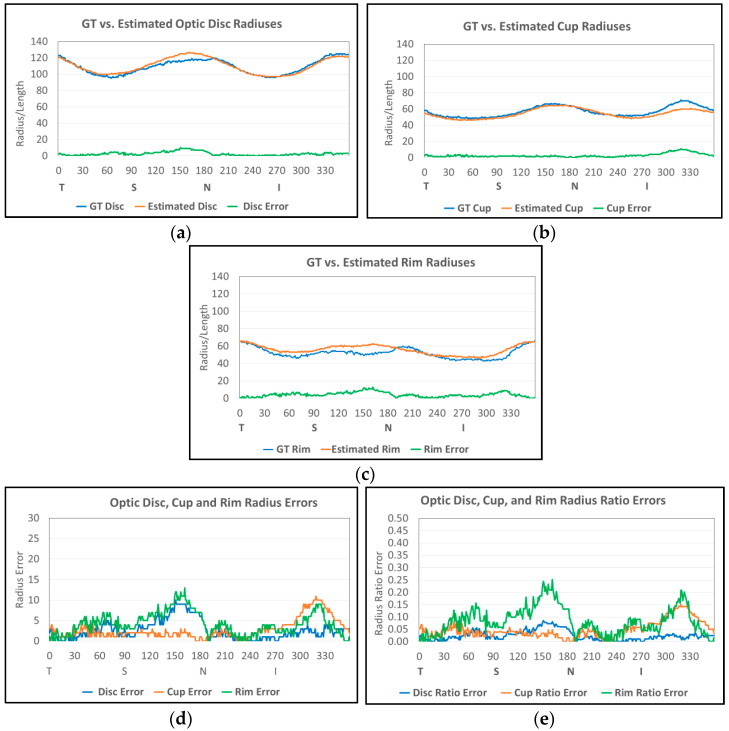
Graphs of optic disc, cup and neuroretinal rim radiuses of the image in Figure 6. (**a**) GT and estimated optic disc radiuses and its error. (**b**) GT and estimated cup radiuses and its error. (**c**) GT and estimated neuroretinal rim radiuses and its error. (**d**) Estimated optic disc, cup, and neuroretinal rim radius errors. (**e**) Estimated optic disc, cup, and neuroretinal rim radius ratio errors.

**Figure 8 diagnostics-12-01063-f008:**
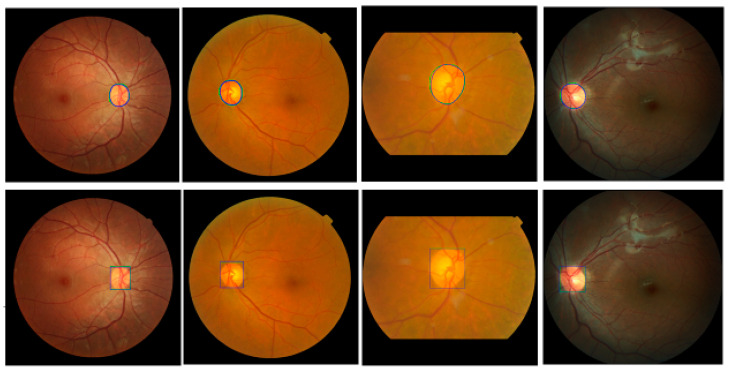
Examples of the proposed Mask R-CNN Results. Top row shows output of optic disc boundaries. Bottom row shows output of ROIs. Green is ground-truth data and blue is estimated Mask R-CNN results.

**Figure 9 diagnostics-12-01063-f009:**
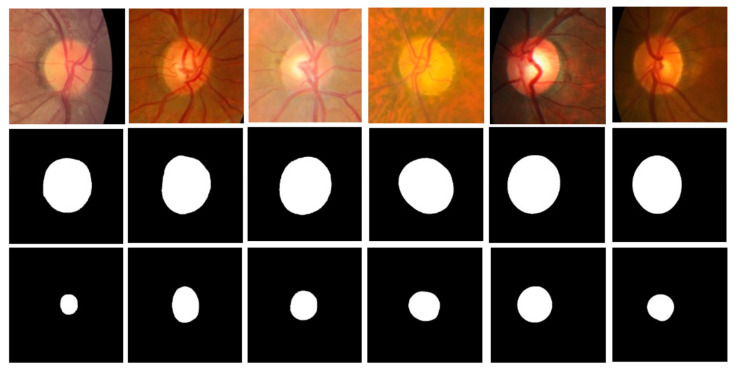
Input and ground-truth ROI images used for training MAPNet. The first row is for ROI images with ROI size = 2.00, the second row is for ground truth label for optic disc, and the third row is for ground truth label for cup.

**Figure 10 diagnostics-12-01063-f010:**
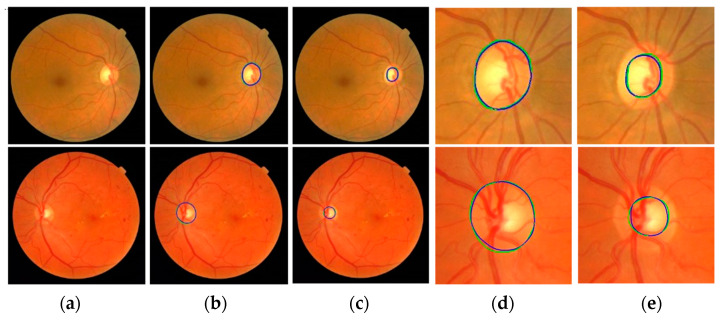
Segmentation output of the proposed method. (**a**) Input image. (**b**) Output of the proposed method for optic disc segmentation. (**c**) Output of the proposed method for cup segmentation. (**d**) Magnified image of the optic disc output (**b**). (**e**) Magnified image of the cup segmentation output (**c**). Green is ground truth and blue is the proposed output.

**Figure 11 diagnostics-12-01063-f011:**
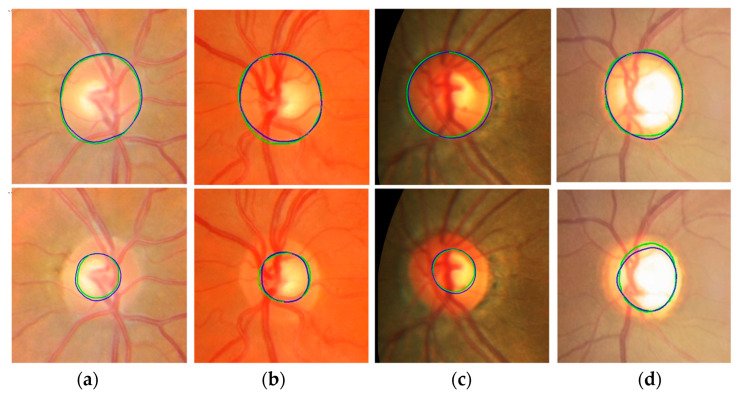
Optic disc and cup segmentation results of the proposed method. The first row is optic disc segmentation results, and the second row is cup segmentation results. In the first and second rows, the two images in the same column are results from the same fundus images. Images in the columns (**a**,**g**) are from Magrabi dataset, images in the columns (**b**,**f**) are from MESSIDOR dataset, images in the columns (**c**,**e**) are from REFUGE dataset, and images in the columns (**d**,**h**) are from Bin Rushed dataset. Green means ground truth and blue means the result from the proposed method.

**Figure 12 diagnostics-12-01063-f012:**
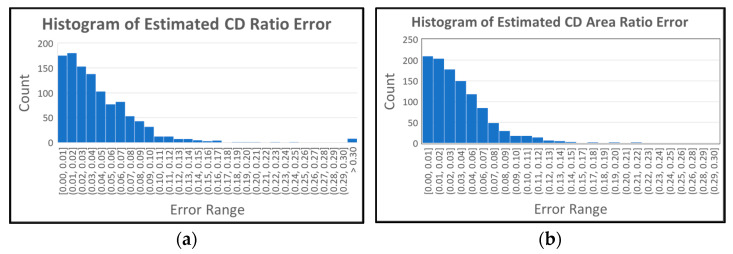
Histograms of estimated CD ration and CD area ratio error. (**a**) Histogram of estimated CD ratio error. (**b**) Histogram of estimated CD area ratio error.

**Figure 13 diagnostics-12-01063-f013:**
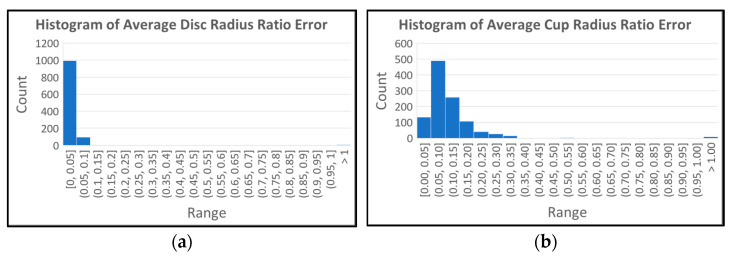
Histograms of average optic disc, cup and rim radius ratio errors. (**a**) Histogram of average optic disc radius ratio errors. (**b**) Histogram of average cup radius ratio errors. (**c**) Histogram of average rim radius ratio errors.

**Table 1 diagnostics-12-01063-t001:** Two datasets used in the experiments.

Dataset	Name	Number of Image	Resolution
RIGA	MESSIDOR	460	2240 × 1488, 1440 × 960
	Bin Rushed	145	2376 × 1584
	Magrabi	94	2376 × 1584
REFUGE	Training set	400	2124 × 2056

**Table 2 diagnostics-12-01063-t002:** Comparison of optic disc segmentation results with different ROI sizes using the proposed MAPNet.

ROI Size	Jaccard Index	Dice Coefficient	Sensitivity	Specificity	Accuracy
1.25	0.9319	0.9644	0.9656	0.9715	0.9670
1.50	0.9327	0.9648	0.9693	0.9824	0.9773
1.75	0.9330	0.9650	0.9727	0.9874	0.9834
2.00	0.9355	0.9665	0.9763	0.9908	0.9879

**Table 3 diagnostics-12-01063-t003:** Comparison of the proposed optic disc segmentation methods with other methods.

Method	Model	Dataset	Train Images	Test Images	JaccardIndex	Dice
Lim et al.	CNN	MESSIDOR	1200	1200	0.8842	
[12]	[4-fold CV]	SEED-DB	235	235		
Agrawal	U-Shape CNN,	REFUGE &	360	40		0.8800
et al. [18]	Ensemble [1-fold]	DRISHTI-GS	101			
Sevastopol-	Modified U-Net	RIM-ONE	159	159	0.8900	0.9459
sky et al.	[5-fold CV]	DRIONS-DB	110	110		
[20]						
Fu et al.	M-Net, Polar Coor-	ORIGA	325	325	0.9290	
[21]	dinate [1-fold]					
Al-Bander	FC-DenseNet,	ORIGA	455	195	0.8933	0.9421
et al. [22]	Post Processing	DRISHTI-GS		101		
	[1 fold ROI]	RIM-ONE		169		
		DRIONS-DB		110		
		ONHSD		99		
Yu et al.	U-Net based on,	MESSIDO	460		0.9410	0.9694
[23]	Pre-trained	Bin Rushed	195			
	ResNet34	Magrabi		95		
	Post processing	DRISHTI-GS	50	50		
	[1-fold]	RIM-ONE	97	32		
Orlando	U-Net+DeepLabv3	REFUGE	800	400		0.9602
et al. [24]	Mask R-CNN	REFUGE	800	400		0.9464
	U-Net [1-fold]	REFUGE	800	400		0.9525
Kim et al.	Modified U-Net3	RIGA * &	699	699	0.8991	0.9455
[31]	[10-Fold]	REFUGE	400	400		
	Mask R-CNN	RIGA * &	699	699	0.9037	0.9489
	[10-Fold]	REFUGE	400	400		
	Mask R-CNN,	RIGA * &	699	699	0.9234	0.9597
	Modified U-Net2	REFUGE	400	400		
	[10-Fold]					
Proposed	Mask R-CNN,	RIGA * &	699	699	0.9381	0.9679
Method	MAPNet	REFUGE	400	400		
	[10-fold CV]					
Proposed	Mask R-CNN,	RIGA * &	699	699	0.9432	0.9707
Method	MAPNet	REFUGE	400	400		
	[Best Set]					

* RIGA includes MESSIDOR, Bin Rushed, and Magrabi datasets. CV means cross validation. Jaccard = Jaccard Index and Dice = Dice Coefficient.

**Table 4 diagnostics-12-01063-t004:** Comparison of the proposed cup segmentation methods with other methods.

Method	Model	Dataset	Train Images	Test Images	Jaccard Index	Dice
Lim et al.	CNN [4-fold CV]	SEED-DB	235	235	0.7698	
[12]						
Agrawal	U-Shape CNN,	REFUGE	360	40		0.6400
et al. [18]	Ensemble Learning	DRISHTI-GS	101			
	[1-fold]					
Sevastopol-	Modified U-Net	RIM-ONE	159	159	0.7044	0.8272
sky et al.	[5-fold CV]	DRISHTI-GS	50	50		
[20]						
Fu et al. [21]	M-Net, Polar Coor-	ORIGA	325	325		0.7700
	dinate [1-fold]					
Al-Bander	FC-DenseNet	ORIGA	455	195	0.6792	0.7939
et al. [22]	Post Processing	DRISHTI-GS		101		
	[1 fold ROI]	RIM-ONE		169		
Yu et al. [23]	U-Net based on	MESSIDOR	460	0	0.7973	0.8725
	Pre trained	Bin-Rushed	195	0		
	ResNet34	Magrabi	0	95		
	Post processing	DRISHTI-GS	50	50		
	[1-fold]	RIM-ONE	97	32		
Orlando	U-Net+DeepLabv3	REFUGE	800	400		0.8826
et al. [24]	Mask R-CNN,	REFUGE	800	400		0.8837
	U-Net [1-fold]	REFUGE	800	400		0.8728
Kim et al.	Modified U-Net3	RIGA * &	699	699	0.7766	0.8694
[31]	[10-Fold]	REFUGE	400	400		
	Mask R-CNN	RIGA * &	699	699	0.7697	0.8617
	[10-Fold]	REFUGE	400	400		
	Mask R-CNN	RIGA * &	699	699	0.7833	0.8742
	Modified U-Net2	REFUGE	400	400		
	[10-Fold]					
Proposed	Mask R-CNN,	RIGA * &	699	699	0.8222	0.8996
Method	MAPNet	REFUGE	400	400		
	[10-fold CV]					
Proposed	Mask R-CNN,	RIGA * &	699	699	0.8355	0.9082
Method	MAPNet	REFUGE	400	400		
	[Best Set]					

* RIGA includes MESSIDOR, Bin Rushed, and Magrabi datasets. CV means cross validation. Jaccard = Jaccard Index and Dice = Dice Coefficient.

**Table 5 diagnostics-12-01063-t005:** Performance of the proposed algorithms based on six measures.

Measure (Mean of the Measure)	Dataset	Test Images	Error
Cup to Disc Ratio Error (CDRE) (Equation (5))	MESSIDO	460	0.0436
	Bin Rushed	145	0.0470
	Megarabi	94	0.0453
	REFUGE	400	0.0460
	Total	1099	0.0451
Cup to Disc Area Ratio Error (CDARE) (Equation (6))	MESSIDOR	460	0.0351
	Bin Rushed	145	0.0333
	Megarabi	94	0.0404
	REFUGE	400	0.0414
	Total	1099	0.0376
Rim to Disc Area Ratio Error (RDARE) (Equation (7))	MESSIDOR	460	0.0351
	Bin Rushed	145	0.0333
	Megarabi	94	0.0404
	REFUGE	400	0.0414
	Total	1099	0.0376
Average Disc Radius Ratio Error (ADRRE) (Equation (11))	MESSIDOR	460	0.0399
	Bin Rushed	145	0.0327
	Megarabi	94	0.0316
	REFUGE	400	0.0722
	Total	1099	0.0500
Average Cup Radius Ratio Error (ACRRE) (Equation (12))	MESSIDOR	460	0.1048
	Bin Rushed	145	03.370
	Megarabi	94	0.2910
	REFUGE	400	0.2970
	Total	1099	0.2257
Average Rim Radius Ratio Error (ARRRE) (Equation (13))	MESSIDOR	46	0.1589
	Bin Rushed	145	0.3433
	Megarabi	94	0.3501
	REFUGE	400	0.2056
	Total	1099	0.2166

## Data Availability

The data used to support the findings of this study are available on request from the following links: https://refuge.grand-challenge.org/details/request (for REFUGE) (accessed on 10 September 2018) and https://deepblue.lib.umich.edu/data/concern/data_sets/ 3b591905z (for RIGA) (accessed on 3 May 2018).

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
