# Peer review of "Identifying Those at Risk of Glaucoma: A Deep Learning Approach for Optic Disc and Cup Segmentation and Their Boundary Analysis"

_diagnostics, 2022, doi:10.3390/diagnostics12051063_

Round 1

Reviewer 1 Report

This article describes a new proposed automated technique for extracting key features, such as cup to optic disc (CD) ratio, CD area, neuroretinal rim to optic disc area ratio, and rim thickness from retinal fundus images. The approach achieves good accuracy and has potential to be incorporated into clinical workflow.  

The article would be improved by enhancing the motivation and expanding description of the unmet clinical need addressed by this automated approach (lines 121-126).  Also, improving figure quality, especially figures 1, 3, and 7 would greatly enhance ability to understand methods and results.  Figures 12 and 13 could also be made larger.

Novelty/added-value of proposed MAPNet architecture could be emphasized especially in lines 292-294 following from previous paragraph.

Use of multiple image datasets and cross-fold validation and data augmentation addresses the small-to-moderate dataset size of 1099 images.  Use of multiple metrics strengthens description of results and conclusions.  Perhaps beyond the scope of this paper, but it would be interesting to see the impact on screening time/accuracy resulting from using the calculations arrived at via this automated approach.  

Author Response

1. The article would be improved by enhancing the motivation and expanding description of the unmet clinical need addressed by this automated approach (lines 121-126).

=>I added a paragraph in lines 127-134 

2. Also, improving figure quality, especially figures 1, 3, and 7 would greatly enhance ability to understand methods and results.  Figures 12 and 13 could also be made larger.

=>I updated Figures 1, 3, 7, 12, and 13.

3. Novelty/added-value of proposed MAPNet architecture could be emphasized especially in lines 292-294 following from previous paragraph.

=>I added a paraph in lines 304-316.

Use of multiple image datasets and cross-fold validation and data augmentation addresses the small-to-moderate dataset size of 1099 images. Use of multiple metrics strengthens description of results and conclusions

4. Perhaps beyond the scope of this paper, but it would be interesting to see the impact on screening time/accuracy resulting from using the calculations arrived at via this automated approach.

=> I added a paragraph in lines 802-813.

Reviewer 2 Report

The manuscript proposes an automatic method to segment optic disc and cup from fundus images using deep learning algorithms. The authors also demonstrated the effectiveness in the segmentation of optic disc and cup when compared with other state-of-the-art methods and provided several 
key measures to screen potential glaucomatous damage

My comment

- Figure 3 for proposed algorithm: parts of the figure are not clear

- Figure 12, 13: revise y axis, the axis title should be separated from values

Author Response

1. Figure 3 for proposed algorithm: parts of the figure are not clear

=> I updated the figure.

2.  Figure 12, 13: revise y axis, the axis title should be separated from values

=> I updated the figures.